# OpenReview forum: "Inference-time Alignment of LLMs at the Token Level"
_ICLR.cc/2025/Conference — Submitted to ICLR 2025_

### Official Review · Reviewer_cDQV · 2024-10-29

**Soundness:** 3
**Presentation:** 2
**Contribution:** 2
**Rating:** 5
**Confidence:** 5

**Summary:**

LLMs are instruction tuned to align them: to better follow user instructions, safety, improve reasoning, etc. In this paper, the authors propose a training free instruction fine tuning approach, where they modify the output sequence of a large base model (not aligned), using the output sequence of a small instruction tuned model, to align the big model. Specifically, whenever the distributions of the small and the big model do not match, they append the big model’s output with the nudging token i.e. the token as per the small aligned model. The empirical results and quite good and convincing, and the approach is definitely novel to the best of my knowledge. However, the paper misses a discussion on key limitations of the proposed approach and a proper discussion and providing intuition behind various sections critical to the functioning of the proposed approach.

**Strengths:**

- Quite an interesting idea and approach, with clean and convincing results.
- Empirical results are done on both standard and safely alignment benchmarks.

**Weaknesses:**

- Many task capabilities come with scale (emergence https://arxiv.org/abs/2206.07682). I fail to understand how the partial completion (generated by the bigger model) when fed to the smaller instruction tuned model would be comprehended in these cases. Can authors test their approach on some of these tasks, with emergence properties (i.e. performance is 0 below a param size, and jumps suddenly at a bigger scale)?
- I fail to understand the argument in L191, i.e. the distribution at alignment related positions is the same in models of different sizes. Aligned models of different sizes, at same numerical position (say the 2nd position where the score is below threshold), might be producing totally different outputs (e.g., small model might be towards the end of solution, bigger model might just be at the start). I fail to understand that the steering tokens generated at these positions are the same or have the desired effect on the big model.
- Lack of discussion on limitations of the approach: This approach inherently assumes that all the mis-alignment occurs at the token level, and does not add a discussion or a section of limitations. For example, the approach won’t work when the misalignment is due to lack of long  context dependencies or sub-sentence contradictions (which is fine t0 have). Would be great if authors can address the same.
- I am not sure about how much this approach will be used in practice. LLM instruction finetuning is not a very costly process (for folks who pretrained a model). If someone in practice really wants a reliable instruction tuned model, with specific capabilities, I doubt they will use this training free approach. For example, one cannot really “safety align” a model using this approach, as it will be quite easy to break, by attacking the small instruction tuned model to mimic distribution of big base model, etc.
- I would really encourage the authors to spend a page or so in their next version, on practical implications/usecases/discussion of their approach. How can this approach guide better instruction tuning or other insights the authors might have.
- Finally, the ICLR author guidelines require an ethics statement and a reproducibility statement. I would encourage the authors to follow the guidelines (I WILL NOT be considering this for my score though, just pointing it out).

**Questions:**

See the weakness section.
Overall, I would recommend the authors that in such papers, where you are proposing a “supposedly” counterintuitive algorithms, focusing on discussion/providing intuition into the results is crucial, and could really improve the work.
I would be willing to increase the score, if authors clarify the intuition/address the questions I posed above (mainly related to why L191, task capabilities).

---

> ### Author Response · Authors · 2024-11-22
> **Response to Reviewer cDQV**
>
> We thank the reviewer for the detailed and helpful comments. We address the reviewer’s concerns below.
>
> We kindly request the reviewers to consider whether our clarifications and added discussions adequately address their concerns, and if so, we would greatly appreciate reconsideration of the scores to better reflect the revised assessment.
>
> **We highlighted the changes and the added new content in our latest submission to make it easier for the reviewers to track them. We will remove the highlights at the end of the discussion session.**
>
> ## [W1] How can a small nudging model effectively nudge a large base models on tasks that requires emergent abilities?
> It is true that many task capabilities come with scale. In fact, many of the tasks we used, such as MMLU and GSM8K, are used to show the emergent abilities [1], and nudging works well on these datasets. One of the main insights from our work is that we can largely disentangle the alignment-related formatting abilities simply at the token level from the general task capabilities (such as many of the emergence capabilities). As illustrated in Table 7, as well as in the examples in the appendix, the nudging tokens are mostly transition or formatting tokens such as “1.” or “Great”, and our scaling-up analysis (section 4.5) suggests that we don’t need a very large nudging model to effectively generate these formatting tokens.
>
> It is indeed interesting that even when the small aligned model does not have strong task capabilities it can still generate nudging tokens that guide the more capable base model to a good answer. Take the last-letter-concat (LLC) dataset for an example of the emergent abilities: for both Llama-2 and Gemma-2 models, the small chat models can only solve around 5% of the questions, while the large chat models perform significantly better (Table 3). As shown in Table 3, even themselves not able to perform the task, the small chat models can still effectively nudge the large base models. One analogy to understand this is to consider how humans write code using CoPilot, where we usually don’t deeply understand how to use each library, but simply through writing comments and defining proper variables and function names we can guide the CoPilot model to generate the code we want.
>
> ## [W2] Clarification of the setups in Section 2.2 (Line 191)
> We are sorry for the confusion about the experiment setups. The goal of section 2.2 is to show that at alignment-related positions, the aligned models of different sizes tend to have similar token distributions. This is verified in the following way: **given the same prefix and generating the next token** for a large base, a large aligned, and a small aligned model, when the large base and large chat disagree about the next token, we measure how often do the small chat and large chat models agree. Similar to Section 2.1, we compare the token distributions of three models on the large aligned models’ answers. As an example, consider the prefix: “Question: What is the smallest breed of Dog? The smallest breed of dog is Chiwawa.”. The next token generated by the large base model can be “Question”, where the base model follows the format and tries to generate another question. On the other hand, both a small and large chat model generate “On average …” to elaborate on their answers. So the next token position is alignment-related since the large base and large chat disagree, but the small chat and large chat agree on what to generate next. We have elaborated the setup in the paper.
>
> ## [W3] Limitation section
> We thank the reviewer for the suggestions of potential limitations of nudging. We have expanded Section 7, describing the potential limitations and future directions of this approach, such as the inference latency of the API-based implementation and learning a nudging model to better model customized alignment rules.
>
> ## [W4&W5] Application of nudging in practice.
> We thank the reviewer for pointing this out. We have addressed this concern in the general response.
>
> ## [W6] Ethics and reproducibility statement.
> Our work does not involve topics mentioned in the ICLR 2025 Author Guide such as “studies that involve human subjects or practices to data set releases”. Still, we have added an ethics and reproducibility statement to our paper. We thank the reviewer for the suggestion.
>
> [1] Wei J, Tay Y, Bommasani R, et al. Emergent abilities of large language models[J]. arXiv preprint arXiv:2206.07682, 2022.

---

> > ### Comment · Reviewer_cDQV · 2024-11-24
> >
> > My concerns continue to remain there. Is nudging only for “alignment” related instruction fine tuning?
> > Authors mention that nudging can improve performance on GSM8k. Does it mean reasoning can be improved only by adding tokens like “Think” or “Great” or other formatting tokens? Does nudging work on say GSM8k even when prompting base model with CoT and the standard fewshot GSM8k evaluation template? Finally, repeating the limitations question I asked originally . What about  cases where instruction finetuning is done to instill capabilities like longer context, add more reasoning capabilities to base model, coding skills, etc.

---

> ### Author Response · Authors · 2024-11-25
>
> > Is nudging only for “alignment” related instruction fine tuning? Authors mention that nudging can improve performance on GSM8k. Does it mean reasoning can be improved only by adding tokens like “Think” or “Great” or other formatting tokens?
>
> We thank the reviewer for the great and deep question! One way to understand that the aligned models have a much better zero-shot performance on reasoning tasks like GSM8K is that after alignment, these models learn to elaborate on their reasoning steps (CoT) before outputting the final answer. Nudging and (few-shot) CoT prompting help the base models similar to this. Similar to (few-shot) CoT prompting, nudging leverages specific answer formatting to elicit the model’s latent reasoning abilities, rather than directly enhancing those abilities. But different from prompting, nudging more dynamically guides the base model to adopt a step-by-step CoT-style reasoning process by injecting tokens such as “Step 1,” “First,” or “1.,” at proper positions during inference. Another key insight here from nudging is that carefully selecting certain answer tokens (e.g., tokens with high uncertainty) can significantly improve performance on reasoning tasks. Recent work has also explored this idea [1].
>
> > Does nudging work on say GSM8k even when prompting base model with CoT and the standard fewshot GSM8k evaluation template?
>
> **We want to emphasize that one of the main goals of RLHF/instruction tuning is to get away from the few-shot examples since people seldom provide few-shot demonstrations when query these models with questions in practice. As a result, we mainly focus on the zero-shot setting in this paper, and comparing with few-shot is not consistent with the motivations and main arguments in our paper.** Still, to partially address the reviewer's concern, we follow the reviewer’s suggestion and test the few-shot performance of Gemma-2-27b on GSM8K. We use the 8-shot CoT prompt from [2]. The 8-shot cot (78.2) has a similar performance to zero-shot nudging (74.6). This suggests that nudging can unlock the base model’s abilities similar to in-context learning but without requiring demonstrations. We want to point out that coming up with a setting to comprehensively and fairly compare these two methods is non-trivial. Such a comparison usually leads to a whole new paper [3]. On the other hand, as both nudging and few-shot prompting work by formatting the model’s answer, how to effectively combine these two paradigms at inference time is a deep question and requires future exploration.
>
> > Finally, repeating the limitations question I asked originally. What about cases where instruction finetuning is done to instill capabilities like longer context, add more reasoning capabilities to base model, coding skills, etc.
>
> We agree that extensive post-training can indeed instill new capabilities, such as longer context or more reasoning capabilities. We find instructions that are beyond the inherent abilities of the base model are usually also challenging for nudging. For example, consider the instruction from the just-eval-instruct dataset: "Write a sentence with all words starting with the letter 'Y' to praise me." We find this instruction to be challenging for nudging because the base models lack the capability to follow such a specific and challenging instruction (see more in our newly added discussion in Appendix E.1). Notably, none of the large aligned models can generate a correct answer for this instruction either. To conclude, we acknowledge that nudging, while simple and training-free, is not a comprehensive solution yet and is constrained by the underlying capabilities of the base model (we have added this discussion to the limitation section). Nevertheless, we believe it represents a step in the broader exploration of techniques to adjust model behavior at the token level during inference. We are excited to investigate ways to enhance nudging in the future to address such limitations.
>
> [1] https://github.com/xjdr-alt/entropix/tree/main/entropix
>
> [2] Wei J, Wang X, Schuurmans D, et al. Chain-of-thought prompting elicits reasoning in large language models[J]. Advances in neural information processing systems, 2022, 35: 24824-24837.
>
> [3] Mosbach M, Pimentel T, Ravfogel S, et al. Few-shot fine-tuning vs. in-context learning: A fair comparison and evaluation[J]. arXiv preprint arXiv:2305.16938, 2023.

---

> > ### Comment · Reviewer_cDQV · 2024-11-28
> >
> > I thank the authors for the experiments on nudging vs fewshot performance on Gemma (base model). From the results, it indeed seems that fewshot demonstrations in the context can perform better / give performance similar to nudging.
> > - Authors mention that goal of RLHF is to get away from fewshot demonstrations. However, the fewshot results also question many other motivations of the paper like low cost evaluations during pretraining, etc. mentioned in the general response. This is because all those goals can simply be accomplished by evaluations with few-shot prompting.
> > - I also feel that if you prompt the model: "Think step by step or so" even without explicit fewshot examples, the performance will be close to that of with nudging.
> >
> > I am inclined towards maintaining my evaluation score.

---

> ### Author Response · Authors · 2024-11-30
> **Clarification of the motivation and evaluation setup of our work.**
>
> Nudging brought up an interesting insight that reasoning abilities can be elicited simply by adding a few tokens during decoding, and we have shown that nudging (zero-shot) can match the performance on reasoning tasks with **manually and carefully constructed few-shot examples** [1]. **Despite this, we’d like to clarify that comparing with few-shot prompting on reasoning tasks is not the main focus of our work.** As indicated by the title of our paper, our primary focus is on alignment, where the model needs to respond to user queries that are highly diverse in topics and language [2, 3]. In such scenarios, providing carefully crafted few-shot examples for every possible user query is impractical or even impossible. As a result, most recent works on alignment/RLHF/safety evaluate models in a zero-shot setting [3, 4, 5, 6, 7]. Following these works, we focus on the zero-shot evaluation setup.
>
> In general, while few-shot learning has demonstrated powerful capabilities, zero-shot abilities are equally important [8, 9, 10, 11, 12]. Zero-shot abilities enable models to generalize to new tasks without relying on task-specific examples, which may not be available or practical in real-world applications, reducing potential computational overheads and avoiding issues like prompt variability [13, 14, 15].
>
> Regarding your point about prompting the model with phrases like "Think step by step", we acknowledge that such prompts can enhance reasoning to some extent [8]. However, our method focuses on a more general approach to alignment that does not rely on specific prompt engineering, which may not generalize across all tasks and user inputs.
>
> We hope we have clarified the motivation of our work, and are willing to discuss further if the reviewer has any more questions.
>
> [1] Wei J, Wang X, Schuurmans D, et al. Chain-of-thought prompting elicits reasoning in large language models[J]. Advances in neural information processing systems, 2022, 35: 24824-24837.
>
> [2] Zhao W, Ren X, Hessel J, et al. Wildchat: 1m chatGPT interaction logs in the wild[J]. arXiv preprint arXiv:2405.01470, 2024.
>
> [3] Cui G, Yuan L, Ding N, et al. ULTRAFEEDBACK: Boosting Language Models with Scaled AI Feedback[C]//Forty-first International Conference on Machine Learning. 2024.
>
> [4] Zheng L, Chiang W L, Sheng Y, et al. Judging llm-as-a-judge with mt-bench and chatbot arena[J]. Advances in Neural Information Processing Systems, 2023, 36: 46595-46623.
>
> [5] Zhou C, Liu P, Xu P, et al. Lima: Less is more for alignment[J]. Advances in Neural Information Processing Systems, 2024, 36.
>
> [6] Dubois Y, Galambosi B, Liang P, et al. Length-controlled alpacaeval: A simple way to debias automatic evaluators[J]. arXiv preprint arXiv:2404.04475, 2024.
>
> [7] Ghosh S, Evuru C K R, Kumar S, et al. A Closer Look at the Limitations of Instruction Tuning[J]. arXiv preprint arXiv:2402.05119, 2024.
>
> [8] Kojima T, Gu S S, Reid M, et al. Large language models are zero-shot reasoners[J]. Advances in neural information processing systems, 2022, 35: 22199-22213.
>
> [9] Wei J, Bosma M, Zhao V Y, et al. Finetuned language models are zero-shot learners[J]. arXiv preprint arXiv:2109.01652, 2021.
> [10] Sanh V, Webson A, Raffel C, et al. Multitask prompted training enables zero-shot task generalization[J]. arXiv preprint arXiv:2110.08207, 2021.
>
> [11] Chung H W, Hou L, Longpre S, et al. Scaling instruction-finetuned language models[J]. Journal of Machine Learning Research, 2024, 25(70): 1-53.
>
> [12] Achiam J, Adler S, Agarwal S, et al. Gpt-4 technical report[J]. arXiv preprint arXiv:2303.08774, 2023.
>
> [13] Zhao Z, Wallace E, Feng S, et al. Calibrate before use: Improving few-shot performance of language models[C]//International conference on machine learning. PMLR, 2021: 12697-12706.
>
> [14] Liu J, Shen D, Zhang Y, et al. What Makes Good In-Context Examples for GPT-$3 $?[J]. arXiv preprint arXiv:2101.06804, 2021.
>
> [15] Lu Y, Bartolo M, Moore A, et al. Fantastically ordered prompts and where to find them: Overcoming few-shot prompt order sensitivity[J]. arXiv preprint arXiv:2104.08786, 2021.

---

### Official Review · Reviewer_fRQL · 2024-10-31

**Soundness:** 3
**Presentation:** 3
**Contribution:** 3
**Rating:** 8
**Confidence:** 4

**Summary:**

The work introduces an algorithm, _nudging_, that allows for aligned generation with a large base unaligned model and a small aligned one, without any additional training. It works by inserting tokens from the small aligned model at positions where the base model is uncertain (probability for the top-1 token is below a predefined threshold), with the remaining tokens being generated by the base model.

The algorithm leads to improvements on downstream tasks when compared to results of the large base model without nudging, approaching results of the aligned version of the large model, and in some cases outperforming it.

The method allows for combination of models from different model families, which differentiates it from other training-free, inference-time alignment methods.

**Strengths:**

The work presents a novel approach for training-free inference-time alignment. The method shows strong empirical results, with performance improvements over the base model and sometimes exceeding the aligned version. It provides a clear and intuitive algorithm that is easy to understand and implement and enables cross-family model collaboration. The approach has significant potential impact, possibly offering an alternative to standard instruction tuning approaches.

**Weaknesses:**

I don't see any major issues with the paper, there are only a few minor ones related to presentation, mentioned in the the _Questions_ rubric (see the _suggestions_).

**Questions:**

Question:
- Line 224, "_This leads to better performance and facilitates better collaboration between model families with different tokenizers._" - What is the difference in performance?

Suggestions:
- Line 353: The difference should be expressed in percentage points (pp) rather than %.
- Figure 4: The lines are difficult to read without extreme zooming. Consider using solid lines and rescaling the plot for better readability.
- Line 499: Shouldn't "unturned small models" be "untuned small models"?

---

> ### Author Response · Authors · 2024-11-22
> **Response to Reviewer fRQL**
>
> We thank the reviewer for highlighting the importance and potential impact of our work.
>
> ## [Q1] The benefits of using nudging “words” instead of nudging “tokens”.
> We observed that the performance drop on several datasets was significant (over 10% absolute for LLama-2 on GSM8K for example) in our early experiments, and we did not explore this more in-depth. The intuition is that the model thinks it's at token boundaries more often than it should, leading to deviations from the right answer. As an example, for LLama-2 models on GSM8K, the nudging model mostly starts the answer with “Sure”, and the base model completes the word with “ly”, ending up with “Surely”, which usually leads to worse answers (See Appendix 1.1 for more discussion). Also, we demonstrated in Section 4.4 how using nudging “words” enables completely different model families (with different tokenizations of words) to collaborate effectively.
>
> ## [S1] Suggestions and typos.
> We thank the reviewer for the suggestions. We have modified the paper accordingly.

---

> > ### Comment · Reviewer_fRQL · 2024-11-23
> >
> > Thank you for addressing my question. I maintain my original score of 8 and recommend acceptance of the paper.

---

### Official Review · Reviewer_ttgc · 2024-11-04

**Soundness:** 3
**Presentation:** 3
**Contribution:** 3
**Rating:** 6
**Confidence:** 4

**Summary:**

The paper presents a method to align a large base LLM at inference time with the help of a small aligned LLM. Based on the observation that base LLMs are uncertain when generating alignment-related tokens, the authors propose injecting token(s) generated by a small aligned LLM when top-1 token probability assigned by the large base LLM is less than some threshold. The effectiveness of this method is demonstrated on 3 model families and several diverse tasks, across which “nudging” is shown to be comparable to, or sometimes better than, a large aligned LLM.

**Strengths:**

- This paper proposes a simple and intuitive approach for inference-time alignment significantly reducing the computation burden required to align a family of base LLMs.

- The authors provide strong and clear motivation for their approach in section 2, demonstrating experimentally where and when to inject nudging tokens.

- Extensive evaluation and analysis on standard and alignment-related benchmarks are easy to follow and demonstrate the effectiveness of the proposed approach. Additional experimentation, particularly the section on scaling, provide great insight.

**Weaknesses:**

- The topic of computational efficiency is not sufficiently addressed. For one, in table 2, it’s misleading to compare the number of calls to the large base models when each call with nudging generates significantly more tokens (16 vs 1 token) relative to the comparison methods. Moreover, no analysis is provided regarding the frequency at which tokens generated by the large aligned model in each call are discarded due to injecting the nudged token(s), but this analysis is crucial to understand the computational implications of this approach.

- The results on standard benchmarks (section 4.2) suggest that nudging underperforms the large aligned model on most datasets. The slight increase in average performance of nudging seems primarily due to performance increases on a few toy tasks (e.g., coin flip, last letter concatenation).

- Section 4.4 on the collaboration of models from different families is hard to follow and underdeveloped. For one, the results reported in table 6 don’t match those in table 3 (e.g., Gemma-2-27b scoring 16.5 vs 17.7 on MMLU). Moreover, it’s strange that Gemma-2-27b-it results and Gemma-2-27b nudged by Gemma-2-2b-it are not included in table 6, but it seems like nudging Gemma-2-27b with OLMo-7b-it and Llama-2-7b-chat can significantly underperform (e.g., Gemma-2-27b nudged by Gemma-2-2b-it scores 75.6 on GMS8K while nudging with OLMo-7b-it and Llama-2-7b-chat score only 41.0 and 65.0 respectively) despite Gemma-2-2b-it being a much smaller mode (2b vs 7b).

**Questions:**

- On average, of the 16 tokens generated in each call to the large base model, how many are discarded? Why is the generation length chosen as 16 for both the small aligned model and large base model? Can you comment on the effect of the generation length on computational efficiency? If you’re able to cache prefixes/intermediate generations at each step, why not use a generation length of 1 for the large base model and avoid discarding tokens?

- Can you provide intuition for why certain tasks (e.g., coin flip, last letter concatenation) benefit so significantly by nudging relative to the other tasks?  Why do models from the OLMo family perform so poorly on last letter concatenation?

- Are results in table 6 reported on smaller samples of GSM8K and MMLU than table 2? Can you provide some insight into why nudging Gemma-2-27b with OLMo-7b-it and Llama-2-7b-chat underperforms nudging Gemma-2-27b with Gemma-2-2b-it? Is Gemma-2-2b-it a strong model, or is some underlying distributional shift between model families to blame?

- In section 4.3, it’s said that “nudging Llama-2-70b-chat with Llama-2-7b-chat is rated slightly higher than even in helpfulness and engagement over Llama-2-7b-chat, even though over 85% of the tokens are from the base model”. What is the point of nudging one aligned model with another aligned model? What’s the takeaway from this experiment? Why is it interesting that Llama-2-70b-chat nudged by Llama-2-7b-chat is rated higher than Llama-2-7b-chat?

- Can you comment on the interoperability of nudging with non-greedy sampling techniques (e.g., temperature > 0) which are commonly the default for larger aligned models (e.g., GPT-4)?

- In line 6 of Algorithm 1, shouldn’t the operator be flipped i.e., $\text{top-1}(f_B(q, a, c_{<i}^B)) < \gamma$ instead of $\text{top-1}(f_B(q, a, c_{<i}^B)) > \gamma$?

I am willing to raise my score if my questions/concerns are adequately addressed.

---

> ### Author Response · Authors · 2024-11-22
> **Response to Reviewer ttgc (Part 1)**
>
> We thank the reviewer for the positive and constructive feedback. We appreciate the reviewer for recognizing that our results are easy to follow and providing great insight. We address the reviewer’s concerns below.
>
> We kindly request the reviewers to consider whether our clarifications and additional experiments adequately address their concerns, and if so, we would greatly appreciate reconsideration of the scores to better reflect the revised assessment.
>
> **We highlighted the changes and the added new content in our latest submission to make it easier for the reviewers to track them. We will remove the highlights at the end of the discussion session.**
>
> ## [W1&Q1] Analysis of computational efficiency
> We thank the reviewer for bringing up the discussion of computational efficiency. We conduct additional experiments and provide a detailed analysis addressing the reviewer’s concerns, and we have added this discussion to the paper (Appendix C):
> 1. **Computational efficiency comparison with the baselines.** Table 2 only gives a rough comparison of the inference time of different methods. To make a more direct comparison to the baselines, we compare the wall clock running time of nudging and the two baselines: ensemble and proxy-tuning on 100 samples on GSM8K (with Gemma-2 models). As shown in the following table, nudging is nearly 10x faster than the ensemble and 18x faster than the proxy-tuning, both of which require calling the base model for every generated token. Although nudging discards some generated tokens, the wall clock running time results suggest that the number of API calls is the most important factor for computational efficiency, since for the later tokens in the answer, every API call needs to reprocess the full context. By making significantly fewer API calls to the base model, nudging achieves a much faster inference speed than the baselines.
>       |                         | Nudging | Ensemble | Proxy tuning |
>       | ----------------------- | ------- | -------- | ------------ |
>       | wall clock running time | 286s    | 3026s    | 5330s        |
>       | Acceleration            | x1      | x10.6    | x18.6        |
>
> 2. **Analysis of the discarded base token ratios.** The ratio of the number of tokens generated by the base models that are discarded is indeed an important aspect of efficiency. In the following analysis, we focus on the discarded token ratio of the base model, as the nudging model is much smaller and has a minor effect on the inference speed. In the paper, we reported the nudging token ratio in the appendix (Fig. 11). Here we first show that it is strongly connected to the discarded base token ratio, which is defined as the number of discarded base model tokens due to the nudging model divided by the total number of tokens generated by the base model. Assuming in a nudging answer there are $N$ nudging tokens, $B$ base tokens, $T=N+B$ total tokens. The nudging token ratio is therefore defined as $R_N=N/T$. In each nudging round, the nudging model generates 1 nudging token and then the base model continues by generating $L$ completion tokens each time. As a result, there can be at most $L$ base tokens that are discarded in each round. So an upper bound of the discarded token ratio $R_D$ can be derived as
> $$R_D \leq \frac{N * L} {B + N*L} = \frac{L}{R_N^{-1} + L - 1} := R’_D$$
> We calculate the nudging ratios $R_N$, actual discarded base token ratio $R_D$, and our derived upper bound $R'_D$ for 3 model families on the just-eval-instruct dataset. As shown in the following table, we find that this simple upper bound gives a fairly accurate estimate of $R_D$. The $R_D$ with $L=16$ in practice is usually around 50% to 80%, which can be further optimized for efficiency by choosing an $L$ more carefully or using an adaptive $L$. However, we note that for the API-based implementation, the inference time is dominated by the number of API calls. We leave the improvement of efficiency as future work.
>  |        | Lama-2 ($\gamma=0.4$) | Lama-2 ($\gamma=0.3$) | Gemma ($\gamma=0.4$) | Gemma ($\gamma=0.3$) | OLMo ($\gamma=0.4$) | OLMo ($\gamma=0.3$) |
>    | :----- | --------------------- | --------------------- | -------------------- | -------------------- | ------------------- | ------------------- |
>    | $R_N$  | 15.7                  | 11.4                  | 12.7                 | 5.5                  | 23.3                | 17.9                |
>    | $R'_D$ | 74.9                  | 67.3                  | 70.0                 | 48.2                 | 82.9                | 77.7                |
>    | $R_D$  | 73.3                  | 62.4                  | 69.2                 | 47.5                 | 82.0                | 76.0                |

---

> ### Author Response · Authors · 2024-11-22
> **Response to Reviewer ttgc (Part 1 continued)**
>
> ## [W1&Q1] Analysis of computational efficiency (continued)
>
> 3. **How is L=16 chosen?** Having a large $L$ means fewer potential API calls but more discarded tokens, while a small $L$ makes more calls and reduces discarded tokens. A proper $L$ should balance these two, we choose a slightly larger number based on our analysis in Section 2: choosing a threshold that captures 10% - 15% tokens would capture most alignment-related tokens, we choose an $L$ that leads to a nudging token ratio $1/L$ that approximately matches this number and slightly higher to reduce the potential API calls. We didn’t optimize heavily on this.
> 4. **Choosing L for caching-based implementation.** A speculative decoding style implementation that caches the question, as well as the generated token, should largely improve the inference time of nudging, and we will release this with the final version. In this case, indeed we can generate only 1 token at a time to avoid discarding tokens. We thank the reviewer for the constructive feedback and will take this into consideration in our implementation while balancing other potential computational overheads.

---

> ### Author Response · Authors · 2024-11-22
> **Response to Reviewer ttgc (Part 2)**
>
> ## [W2] Nudging underperforms the large aligned model on most datasets.
> We want to point out that nudging is a simple and training-free inference-time method, and large aligned models are not one of our baselines to beat (see more discussion in the general response). Instead, we want to demonstrate that without explicit alignment training, nudging a large base model with a much smaller aligned model can achieve performance close to a large aligned model. The strong performance of nudging on math and symbolic datasets shows a particular type of task that nudging would potentially benefit from.
>
> ## [W3&Q3] Improving the clarity of Section 4.4
> We appreciate the review for pointing this out. It is correct that the results in Section 4.4 are conducted on smaller subsets (200 samples for each dataset). We apologize for the confusion and have already updated the full dataset results in the latest submission. In the following, we address the reviewer’s concerns individually:
> - **The motivation.** The main goal of this section is to demonstrate that different from previous model ensembling methods, e.g., proxy-tuning, nudging allows to combine models from different families. This can be potentially useful in many scenarios. For example, when a powerful series of base models comes out. One can easily improve the nudging system’s capabilities simply by switching to a better base model. We aim to simulate such a use case using Table 6, where Gemma-2 is the “new” series of base models. Nudging Gemma-2-27b with “existing” small chat models like Llama-2-7b-chat can largely outperform the best “existing” aligned model Llama-2-70b-chat.
> - **Comparison with Gemma-2-2b-it nudging results, and Gemma-2-27b-it results.** As discussed in the previous point, comparing with Gemma-2-2b-it nudging results or the Gemma-2-27b-it results is not the main point of the section. Importantly, despite being a smaller model, Gemma-2-2b-it largely outperforms OLMo-7b-it and LLama-2-7b-chat on most datasets, especially on math datasets. This may explain the observation that nudging with Gemma-2-2b-it significantly outperformed nudging with OLMo-7b-it and LLama-2-7b-chat on GSM8K. Still, on MMLU, nudging with Gemma-2-2b-it (66.8) performs similarly to nudging with OLMo-7b-it (64.4) and LLama-2-7b-chat (67.0). Considering the different pretraining and post-training data distributions and training settings for different model families, using nudging models from the same model families might lead to better performance in general. However, answering this question is beyond the scope of this paper, and we would love to understand this problem in future works.
>
> ## [Q2] Nudging benefits certain tasks.
> The fact that nudging benefits particularly math and symbolic reasoning tasks is intriguing. As discussed in Section 4.2, there are existing works [1] showing that instruction-tuned models can underperform their base versions in factual and reasoning tasks. As an example, we find on the Coin Flip dataset, that both the LLama-2 and Gemma-2 instruction-tuned models tend to answer that the coin has an equal chance of being head or tail up. The base models explore less of such a behavior. By disentangling alignment and pre-training, nudging can potentially mitigate some countereffects introduced by the alignment process on certain tasks. However, we don’t have the model training data to accurately attribute this phenomenon.
>
> ## [Q4&Q6] Typos in section 4.3 and Algorithm 1.
> We are sorry for the confusion and have fixed the typos in our latest submission.
>
> ## [Q5] The interoperability of nudging with non-greedy sampling.
> Our code supports different sampling methods for both base and nudging models. For reproducibility and simplicity, we use greedy sampling for all our experiments following previous work [2, 3].
>
> [1] Wang Y, Ivison H, Dasigi P, et al. How far can camels go? exploring the state of instruction tuning on open resources[J]. Advances in Neural Information Processing Systems, 2023, 36: 74764-74786.
>
> [2] Liu A, Han X, Wang Y, et al. Tuning language models by proxy[J]. arXiv preprint arXiv:2401.08565, 2024.
>
> [3] Kojima T, Gu S S, Reid M, et al. Large language models are zero-shot reasoners[J]. Advances in neural information processing systems, 2022, 35: 22199-22213.

---

> > ### Comment · Reviewer_ttgc · 2024-11-22
> >
> > I appreciate the detailed responses to my comments. A number of the concerns I had stem from the implementation of Nudging being very API-focused. Certainly, it's fantastic that Nudging can work over API, but I suspect that the additional cost and time of reprocessing the input after each nudging token + the high ratio of discarded tokens from the base model will limit the practicality of Nudging over API, at least until API providers offer more options for caching between requests. I believe the manuscript would have been stronger had the authors presented a variant of Nudging with L=1 and an efficient caching solution for locally-run models, as I suspect that such an implementation will still exhibit the same strong performance increases as those reported in the manuscript but with higher efficiency.
> >
> > Implementation aside, I believe the underlying algorithm behind Nudging to be novel, creative, and effective. I also agree that it could serve as a solid foundation for future work on inference-time alignment. Because of this, I think this work should be accepted, and have updated my initial score accordingly.
> >
> > Also, I appreciated the discussion on "the practical application and implications" included in your general response. In particular, I think the latter 3 applications should receive more emphasis throughout the manuscript. Related to your last point on personalization, I'm also curious whether Nudging could be used adversarially to align a large base model with a small model trained to produce harmful or unsafe content. If so, I think a thorough discussion on this could be warranted either in the limitations or ethics statement.

---

> > > ### Author Response · Authors · 2024-11-22
> > >
> > > We thank the reviewer for updating the score, and we appreciate the reviewer thinking that "this work should be accepted". We agree that the caching solution for nudging would make the method more practically useful and would release the implementation with the necessary efficiency analysis in our final version.
> > >
> > > We also thank the reviewer for bringing up the potential adversarial use of nudging in producing harmful or unsafe content. We have added the discussion to the ethics statement.
> > >
> > > Once again we thank the reviewer for providing constructive and valuable feedback.

---

### Official Review · Reviewer_hRsS · 2024-11-04

**Soundness:** 3
**Presentation:** 2
**Contribution:** 2
**Rating:** 5
**Confidence:** 3

**Summary:**

The authors propose a method called Nudging, which aims to align large language models (LLMs) during inference without any additional training. Instead of retraining large models for alignment, they use a smaller, aligned model to guide the base model by injecting "nudging tokens" whenever the base model is uncertain about what to generate next. The authors also evaluate Nudging across three model families—Llama-2, Gemma-2, and OLMo—on a set of 13 tasks, including reasoning, general knowledge, instruction following, and safety benchmarks.

**Strengths:**

- The idea of leveraging a small aligned model to steer a larger base model in real-time is  an area that worth studying (e.g., speculative decoding). This paper starts from another perspective and uses the effect of the small model steer model when the model confidence is relatively low. Although some might say the method is very simple, I think it is novel.

- The authors tested Nudging across three different model families (Llama-2, Gemma-2, and OLMo) and 13 diverse tasks. In many cases, Nudging's performance matched or even surpassed that of fully aligned large models, which is good given that no additional training was involved.

- The authors perform analysis in nudging's performances and behaviors both in Section 2 & 4 & 5, which is informative. However, the authors fail to analyze the scenarios that nudging won't work well.

**Weaknesses:**

- Nudging relies on the base model’s uncertainty estimates to decide when to intervene. If the base models themselfe are strong, the effectiveness will decrease (e.g., the results of Gemma 2).

- I don't quite understand the specific practical scenarios for using nudging. In what situations do we need to use nudging, considering that most practical models are already well-aligned, and we will continue to get increasingly powerful base and aligned models? If I'm not mistaken, the improvement nudging offers for well-aligned models shouldn't be very significant.

- The benchmarks selected in the paper are not very challenging these days, and it is kind of weird that the Gemma 2 model performs poorly on the CF task. How will the nudging method perform if we use more challenging benchmarks?

**Questions:**

- Does this method conflict with other methods that enhance performance during the inference stage, such as Chain of Thought and Speculative Decoding?

- The paper focuses primarily on standard benchmarks with shorter completions. It would be interesting to see how well Nudging scales to tasks like essay writing, story generation, or extended dialogue, where maintaining alignment over longer contexts becomes more challenging.

---

> ### Author Response · Authors · 2024-11-22
> **Response to Reviewer hRsS**
>
> We thank the reviewer for the helpful feedback. We are grateful that the reviewer recognizes our method as simple and novel. We address the reviewer’s questions below.
>
> We kindly ask the reviewers to consider whether our responses address their concerns and, if so, to reconsider their scores accordingly.
>
> ## [W1] The effectiveness of nudging would decrease if the base models themselves are strong
> Using a strong base model would increase the performance of nudging, and we view it as an advantage of the modularity of nudging. The motivation for nudging is to disentangle the general and alignment-related capabilities of LLMs. We rely on a small nudging model for formatting the answer and the large base model for general capabilities like factual knowledge and math reasoning. This modularity of nudging allows for improving both sides separately. As Section 4.5 shows, using a stronger base model consistently improves nudging performance across model families and tasks. Also, we show in Section 4.4 that using a stronger base model (Llama-2-70b -> Gemma-2-27b), even if it is from a different model family, can significantly improve nudging performance.
>
> ## [W2] The practical scenarios for using nudging
> We thank the reviewer for bringing up the discussion of the potential applications of our method. We have addressed this concern in the general response.
>
> ## [W3] The benchmarks selected are not very challenging
> We thank the reviewer for the suggestion on evaluating nudging on a wider range of tasks. Nevertheless, in this work, we focus on understanding alignment and proposing an inference-time algorithm for aligning the base LLMs. The benchmarks we used in our work, like GSM8K, MMLU, and Arc-challenge, are widely used in assessing LLMs (both the base and instruction tuned). On these well-studied benchmarks, nudging still gives significant improvements over the large base and small aligned models, and sometimes surpasses the large aligned models’ performance without any training.
>
> For the somewhat synthetic Coin Flip task, the results are also intriguing. We find that in a zero-shot setting, both Llama-2-chat models and Gemma-2-it models tend to answer the coin can either be head or tail up, which aligns with existing works [1] showing that instruction-tuned models can underperform their base versions in factual and reasoning tasks, while nudging addresses the problem (we also discussed the Coin Flip task in Section 4.2, the third paragraph).
>
> ## [Q1] Does this method conflict with other methods that enhance performance during the inference stage, such as Chain of Thought and Speculative Decoding?
> No, chain of thought (CoT) is a prompting method and we can use it in addition to nudging. In some of our experiments, we prompt the model to use CoT by asking it to answer the question by walking through the reasoning steps (see our prompt for reasoning tasks in Figure 10).
>
> For speculative decoding, as we discussed in the discussion section (Section 7), we can implement nudging in a similar way that caches generated tokens during inference to largely reduce inference latency.
>
> ## [Q2] Evaluation nudging on other tasks involving long contexts, such as essay writing and story generation.
> We thank the reviewer for the suggestion, and we are also excited about investigating how nudging works in other tasks in future works. For essay writing and story generation specifically, the Just-Eval-Instruct dataset, which we used to evaluate nudging, already contains relevant questions, such as 'Write an essay discussing the importance of communication in a relationship.” For this example, we checked that the nudging answers are rated high scores of all 5 dimensions by GPT-4o, and nudging performs on par with the chat models on the whole dataset (Section 4.3).
>
> [1] Wang Y, Ivison H, Dasigi P, et al. How far can camels go? exploring the state of instruction tuning on open resources[J]. Advances in Neural Information Processing Systems, 2023, 36: 74764-74786.

---

> ### Author Response · Authors · 2024-11-30
> **PDF modification deadline has passed**
>
> Thank you for your feedback so far. As the deadline for modifying the PDF has passed, we can no longer make updates to the submission. However, we are happy to continue discussing any remaining issues or concerns you may have.

---

### Official Review · Reviewer_HtLx · 2024-11-04

**Soundness:** 3
**Presentation:** 3
**Contribution:** 3
**Rating:** 6
**Confidence:** 4

**Summary:**

This paper proposes NUDGING, an innovative approach for aligning base models at inference time without any additional training. NUDGING is a training-free method that leverages a smaller, pre-aligned nudging model to guide the base model’s output on uncertain tokens, making it an efficient alternative to traditional alignment methods. This approach not only maintains strong alignment but also significantly reduces the computational costs associated with training large models for alignment, thereby improving their practical applicability across diverse tasks.

**Strengths:**

1. This paper presents an innovative method, NUDGING, which introduces a novel, training-free approach for aligning base models at inference time. Unlike traditional alignment methods that require fine-tuning across each base model,  NUDGING leverages a small aligned model to "nudge" a larger, unaligned base model by intervening selectively when the base model displays uncertainty. The proposed method does not require any additional training.
2. This paper conducts comprehensive evaluations across multiple model families and a diverse range of tasks, demonstrating that NUDGING achieves comparable or even superior zero-shot performance relative to fully aligned large models, all without additional training.
3. This paper is clear and well-organized, effectively guiding readers through the motivation, design, and implementation of NUDGING.

**Weaknesses:**

1. Although the authors conduct comprehensive experiments across different model families, I am curious about the effect on llama3. We know that the performance of llama3-instruct is great, so how does NUDGING compare with it?

2. The authors use a fixed uncertainty threshold γ to trigger intervention based on the uncertainty of the underlying model. Although an ablation study was done to demonstrate the choice of threshold, is it possible to have an adaptive threshold? This is just a question and does not affect my judgment on this paper contribution.

3. NUDGING’s design requires multiple back-and-forth interactions between the base and nudging models, which adds latency to the inference process, especially for tasks requiring quick response times. Wondering if the author has a way to avoid this?

**Questions:**

See weaknesses

---

> ### Author Response · Authors · 2024-11-22
> **Response to Reviewer HtLx**
>
> We thank the reviewer for the positive and helpful comments! We address their questions below.
>
> **We highlighted the changes and the added new content in our latest submission to make it easier for the reviewers to track them. We will remove the highlights at the end of the discussion session.**
>
> ## [W1] The effect on Llama-3 models
> We tested Llama 3 on the standard benchmarks. We found that, as in the other three model families, nudging shows a significant improvement over the large base model on most datasets. However, nudging Llama-3-70b with Llama-3-8b-instruct underperforms the small nudging model alone (Llama-3-8b-instruct) on many datasets. As shown in the Llama-3 report [1], the llama-3 model family has a specific post-training process aiming to strengthen various capabilities of the model (math, coding, reasoning, etc). As a result, the small instruction-tuned model has better task-relevant abilities than the base models, explaining why including the base model did not lead to further benefits. We have added the discussion and the results to the paper.
>
> ## [W2] Adaptive threshold
> We thank the reviewer for the suggestion. Using adaptive thresholds is a promising direction to improve the effectiveness and efficiency of nudging. We are excited to explore this in future works.
>
> ## [W3] Extra latency to the inference process
> We agree with the reviewer that extra inference latency is a potential concern for the API-based implementation of nudging. Fortunately, as discussed in Section 7, nudging can be implemented in a speculative decoding [2] manner: we cache the current output for both the base and the nudging model and use it for decoding future tokens instead of making separate API calls. Since the nudging models are usually much smaller than the base models, the inference speed of nudging would be similar to the base model alone. We will include an implementation with our code release.
>
> [1] Dubey A, Jauhri A, Pandey A, et al. The llama 3 herd of models[J]. arXiv preprint arXiv:2407.21783, 2024.
>
> [2] Leviathan Y, Kalman M, Matias Y. Fast inference from transformers via speculative decoding[C]//International Conference on Machine Learning. PMLR, 2023: 19274-19286.

---

> ### Author Response · Authors · 2024-11-30
> **PDF modification deadline has passed**
>
> Thank you for your feedback so far. As the deadline for modifying the PDF has passed, we can no longer make updates to the submission. However, we are happy to continue discussing any remaining issues or concerns you may have.

---

### Author Response · Authors · 2024-11-22
**General response**

We thank the reviewers for their positive and constructive feedback. We appreciate that the reviewers recognize our work as **‘innovative“,’ “simple and intuitive”, with “strong and clear motivation”,** and **“has a significant potential impact, possibly offering an alternative to standard instruction tuning approaches”.** We are also grateful that the reviewers found **the evaluation comprehensive** and **the content well-organized** and **easy to follow.**

# High-level concerns

We mention the reviewer-specific concerns in the individual responses, but here we’ll summarize some high-level concerns.

## [hRsS, cDQV] The practical application and implication.

We have shown that nudging a large base model with a small aligned model achieves on-par and sometimes better performance than the large aligned models. However, replacing or improving upon existing well-aligned models is not the main goal of nudging. Instead, we aim to offer an alternative approach for alignment that would benefit future development and deployment of new models. Here we list some additional potential **applications** of nudging:
1. **Aligning extremely large LLMs.** Even though post-training or alignment has become standard for every model family, aligning extremely large LLMs, such as LLama-3-405B, can be prohibitively expensive for most. Nudging offers a viable alternative alignment solution to directly fine-tuning these extremely large LLMs.
2. **Frequent evaluation of base LLMs during pre-training.** It is important to monitor the general capabilities of the base models at different stages of pre-training. With nudging we can easily assess how well the base models would perform various tasks after alignment without performing alignment training.
3. **Comparing base LLMs in an easy way without fine-tuning.** Nudging can also be used as a training-free tool for comparing base LLMs from the same or different model families, in regards to their inherent capabilities and potential task performance.
4. **Training-free customized alignment.** As the LLMs become widely used in various fields, there is an increasing need for customized models that meet the different needs in alignment, such as personalization. Nudging offers a simple and modularized solution for alignment without accessing the model weights or the distribution over the whole vocabulary. The customized small aligned model could directly work with upgraded future models without updating, whereas the parameter efficient training methods like LoRA or Adapters require training from scratch for every new model update.

**Model collaboration without weight access.** We want to emphasize that compared with other training-free methods, nudging does not require model weights as long as we have the top-1 log-probs. Such flexibility makes nudging very easy to implement and deploy. Additionally, many widely used methods, such as prompting and in-context learning, start from training-free and are later improved. Our work offers a fresh perspective of training-free alignment and token-level model collaboration.

**Insights into alignment.** Finally, one of the main goals of this work is to provide a better understanding of how alignment changes model behavior and to what extent we can align the model behavior at inference time. Our analysis provides such insights into understanding LLM behaviors and can facilitate the development of future alignment and inference-time methods.

# Summary of changes
We’ve updated the submission in response to the reviewers’ concerns and to improve the content quality. Here is a summary of the changes:
- Figure 1: We optimized the illustration and example. The reference texts in the paper are modified accordingly to explain our method better.
- Section 2.2: We elaborated on the experimental setup for clarity.
- Section 4.4: We improve the writing to further motivate the experiments. The original results are based on smaller subsets of the datasets; we have updated them with results from the full dataset.
- Section 4.5: We improved the writing for conciseness.
- Section 7: We added a section on the potential limitations and future directions of nudging.
- Appendix C: we added a discussion of computational efficiency.
- Appendix E: we added a discussion of a case study of a challenging example and the results for Llama-3.
- Typos and suggestions: We have fixed the typos and modified the content based on the reviewer's suggestions. Thanks again for the helpful comments.
- We added an ethics statement and a reproducibility statement

**We highlighted these changes in our latest submission to make it easier for the reviewers to track them. We will remove the highlights at the end of the discussion session.**

---

> ### Author Response · Authors · 2024-11-28
> **Removing change highlights from the PDF**
>
> Since it is almost the end of the final day we were able to modify the PDF, we have removed the highlights that were initially included to indicate changes in our paper. The reviewers can still refer to our summary of modifications in the general response for the changes we made.

---

### Meta-Review · Area_Chair_DZjm · 2024-12-18

**Metareview:**

This paper proposes a simple approach for aligning larger LMs by using the outputs of a smaller LM at certain points during generation to "nudge" the output of the larger LM. They find that this approach align larger LMs across various model families, and outperforms more expensive alternatives such as proxy tuning.

On the positive side, this paper presents an extremely simple approach that is effective in some cases. On the negative side, it is unclear how practical this approach would be in practice (the authors highlight some potential scenarios in the rebuttal, but these scenarios are not actually explored), and the framing of this approach as enabling easier "alignment" seems like it is bit of a stretch (e.g., GSK8K is arguably not really an alignment dataset).

**Additional Comments On Reviewer Discussion:**

The reviewers were generally on the fence, except for Reviewer fRQL who gave a positive score of 8. However, their review seems somewhat low effort, so it was not given a lot of weight. The main weaknesses pointed out by the reviewers include lack of evaluation on harder benchmark, lack of discussion on limitations, and the potential impracticality of the method. Reviewer ttgc updated their score upward during the rebuttal, while the other reviewers chose to maintain their scores (after some engagement).

---

### Decision · Program_Chairs · 2025-01-22

Reject